# Pruning via Ranking (PvR): A unified structured pruning approach

## Abstract

The increase in width and depth has facilitated neural networks to learn from large amounts of data leading to state-of-the-art results in both vision and NLP tasks. In order to democratize such massive networks, it is important to deploy them on resource-limited devices through model compression techniques such as structured pruning. Unfortunately, most pruning methods are tailored towards compressing specific models due to widely differing network architectures for distinct tasks. At the same time, it is desirable for pruning algorithms to generate optimal sub-networks according to user-specified parameter budgets. In this work, we propose Pruning via Ranking (PvR), a novel, global structured pruning approach which generates dense sub-networks that comply with any user-supplied parameter budget. PvR consists of a grouping module and a ranking module that are used to generate smaller networks in terms of both function composition as well as network width for a given dataset. The smaller networks are then trained from scratch instead of being fine-tuned as we empirically demonstrate using a recently proposed model complexity measure that re-initialization after pruning followed by re-training results in better performance. We compare our method against multiple pruning approaches on benchmark datasets, namely, CIFAR10, Tiny ImageNet and IMDB 50K movie reviews, with standard models, namely, VGG16, ResNet34 and Bert-base-uncased. We use both accuracy and model inference latency metrics to evaluate the performance of each approach. The smaller networks proposed by PvR for a range of parameter budgets when trained from scratch outperform all other methods across all datasets and models. In fact, our recommended sub-networks with fewer layers achieve less than $1\%$ test accuracy drop even after pruning $90\%$ of the original model across all networks and datasets while enjoying lower inference latency due to reduced depth.

## 1 Introduction

Pre-trained models are highly over-parameterized networks that learn effective representations from massive amounts of data. These models have become the cornerstone for achieving state-of-the-art results in varied downstream tasks across the computer vision (Ren et al., 2015; Simonyan & Zisserman, 2014; He et al., 2016) and natural language processing (Radford et al., 2018; 2019; Devlin et al., 2018) domains. Despite their incredible success, pre-trained models are difficult to democratize for use on commercial hardware especially on low-end systems due to their immense size which leads to large memory footprint and inference time. Thus, a good amount of recent deep-learning literature is focused on reducing the size of such large models while keeping their performance intact. Some of these methods include network pruning (LeCun et al., 1989; Hassibi & Stork, 1992; Han et al., 2015; Molchanov et al., 2016; Li et al., 2016), low-rank weight approximation (Denton et al., 2014; Lebedev et al., 2014), weight quantization (Courbariaux et al., 2016; Rastegari et al., 2016), and knowledge distillation (Hinton et al., 2015; Romero et al., 2014) among which pruning-based methods have garnered a lot of attention due to their size reduction versus performance trade-off.

Broadly, there are two approaches to network pruning: i) masking the individual weights followed by fine-tuning/re-training or ii) masking the entire neurons/channels of a pre-trained network followed by fine-tuning/re-training. The first method is known as unstructured pruning which results in a sparse model (Frankle & Carbin, 2018) while the second is known as structured pruning which results in dense sub-networks (Li et al., 2016). Sparse models require specialized hardware and

software to achieve speedup and reduced storage costs (Han et al., 2016). On the other hand, dense sub-networks can reduce inference time and storage costs on any currently available hardware.

Most structured pruning methods are tailored towards compressing specific models due to widely differing network architectures for distinct tasks. For example, a number of pruning methods are dedicated to compressing Convolutional Neural Networks (Sui et al., 2021; Luo & Wu, 2020; Li et al., 2022) while others focus on pruning language models (McCarley et al., 2019; Ma et al., 2023; Hou et al., 2020). Recently, Fang et al. (2023) proposed a method to evaluate dependencies within any architecture and generate groups that need to be pruned together. To showcase the generality of their method, they used $\ell_2$ norm-based pruning (Li et al., 2016), a strong structured pruning baseline, to compress several different networks. Nevertheless, very few structured pruning methods exist that can seamlessly work with both vision and NLP specific models (Chen et al., 2021b). At the same time, it is desirable for pruning algorithms to generate optimal sub-networks according to user-specified parameter budgets since one cannot assume the size of the end device where the smaller network is to be deployed.

Keeping these objectives in mind, we propose PvR (Pruning via Ranking), a novel, neuron/filter ranking approach that can be used to prune both vision as well language models. PvR consists of a grouping module and a ranking module where the former component is used to group similar neurons together for faster pruning while the latter component is used to estimate the global importance of the said groups. Once the groups are ranked globally, the least important ones are removed based on a user-supplied parameter budget leaving behind a dense smaller sub-network that is reduced in both depth and parameter count. The resultant dense sub-network is re-initialized and trained from scratch instead of being fine-tuned using the preserved weights. This is motivated by a recently proposed model complexity measure called, Geometric Complexity (GC) (Dherin et al., 2022). We use GC to empirically demonstrate that re-initialization after pruning followed by re-training is a better heuristic. Through extensive evaluations on benchmark datasets, namely CIFAR10 (Krizhevsky et al., 2009), Tiny ImageNet (Le & Yang, 2015), and IMDB 50K movie reviews (Maas et al., 2011), with both vision and language models, namely VGG16 (Simonyan & Zisserman, 2014) with batch normalization, ResNet34 (He et al., 2016) and Bert (Devlin et al., 2018), we show that the dense, shallower sub-networks recommended by PvR significantly outperform all other methods.

### 1.1 OUR CONTRIBUTIONS

- We propose a novel neuron/filter global ranking method, PvR that automatically generates a shallower, dense sub-network for a given dataset, pre-trained model, and parameter budget.

- We use a recently proposed model complexity measure, Geometric Complexity, to empirically demonstrate that re-initializing a structurally pruned network is a better initialization heuristic.

- Finally, the resultant sub-networks from PvR are re-initialized and trained from scratch with their effectiveness being demonstrated through multiple experiments on benchmark datasets in both the vision and NLP domains against distinct pruning schemes in terms of both accuracy as well as model inference latency.

## 2 RELATED WORK

Many structured pruning approaches have been proposed in the past for convolutional networks (He & Xiao, 2023). Weight-dependent methods evaluate the importance of individual filters in convolutional neural networks without using any input data information. Some of the more famous approaches are Norm-based filter pruning methods such as $\ell_1$ (Li et al., 2016) and $\ell_2$ that remove filters having the smallest $\ell_1$ or $\ell_2$ norm, respectively. Recently, Filter Pruning via Geometric Median (FPGM) (He et al., 2019) was proposed which prunes filters that are close to the geometric median of a particular layer. Activation channel pruning methods utilize the output from the convolution operation on input data to determine which filters to remove. HRank (Lin et al., 2020) removes filters corresponding to the lowest average rank of the activations in the current layer. CHannel Independence or CHIP (Sui et al., 2021) removes filters corresponding to the channels having the highest cross-channel correlation in a given layer. ThiNet (Luo et al., 2017) uses a greedy algorithm with a reconstruction error loss to approximate the next layer's activation map. The filters corresponding to the channels that do not contribute to the reconstruction error are pruned. NISP (Yu

et al., 2018) uses an off-the-shelf feature ranking method to determine the neuron importance of the penultimate layer. These scores are then propagated backward to determine the importance of filters in the previous layers. Finally, the filters with the least importance scores are pruned. CURL (Luo & Wu, 2020) masks one channel at a time and uses KL-divergence as the criterion between a model's original output and the masked channel output to determine the importance of the filter corresponding to the masked channel. Based on global sorting, the least important filters from across the model are pruned. Regularization-based methods introduce a regularization term during model training to induce network sparsity. Network Slimming or NS (Liu et al., 2017) prunes filters whose scaling factor in the batch normalization layer on the corresponding channel output is small. Training of such models is done jointly with the scaling factor and channel-level sparsity-induced $\ell_1$ norm. Optimization tool-based methods utilize either the first or second-order Taylor Expansion to approximate the change in loss induced by filter pruning. One of the more famous methods (Molchanov et al., 2019) uses the first-order expansion to derive importance scores for each filter which is defined as the sum of the product of individual filter weights and their corresponding gradient. Other methods such as Random Channel Pruning or RCP (Li et al., 2022) have been introduced as a structured pruning extension of the Lottery Ticket Hypothesis (Frankle & Carbin, 2018). RCP generates multiple network copies by pruning filters randomly. The networks are then trained for a few epochs and a subset of the best-performing models are selected. These models are further trained and finally, the best-performing model is selected.

A number of structured pruning methods have also been proposed for language models (Hou et al., 2020; Zhu et al., 2023; Devlin et al., 2018). DynaBert (Hou et al., 2020) uses first-order estimation to capture neuron importance scores and then generates both width and layer adaptive BERT (Devlin et al., 2018) sub-networks. Fan et al. (2019) prune entire layers from language models by developing a structured dropout technique known as LayerDrop which acts as a regularizer during training but generates shallower sub-networks, on-demand during inference. Voita et al. (2019) prune entire heads of language models using stochastic gates and a differentiable relaxation of the $\ell_0$ penalty. McCarley et al. (2019) introduce additional trainable parameters or masks to different parts (such as attention heads) of pre-trained language models. The mask values are learned via a single forward pass and the least important parts are pruned. CoFi (Xia et al., 2022) introduces masks at different granularities of a language model which allows the method to produce more flexible structures. It also uses a layerwise distillation objective to transfer knowledge from the unpruned network to the pruned one. LLM-Pruner (Ma et al., 2023) finds non-critical structural dependencies in language models and uses first-order information to safely prune non-optimal groups. It is to be noted that even though structured pruning methods such as OTO (Chen et al., 2021a) work with both vision and language models, they are out of the scope of this work because they do not operate in the paradigm of targeted parameter budget.

## 3 PvR: Pruning via Ranking

Pruning via Ranking (PvR) is a fast pruning algorithm that computes the global importance of neurons/filters across a model with the least important neurons/filters being pruned away. PvR not only reduces the number of parameters but also reduces the depth of a network which allows our recommended sub-networks to be much faster during both training and inference than similarly sized networks generated by other pruning methods. PvR is mainly composed of two components, a grouping module that groups similar neurons/filters and a ranking module that ranks the generated groups. It is to be noted that PvR only performs forward passes on the pre-trained model to generate the groups and their corresponding ranks.

### 3.1 Ranking Module

Let $f_\theta$ be an $L$ layer neural network parameterized by $\theta$ where $\theta = \{\theta_1, \theta_2, \cdots, \theta_L\}$. Here, $\theta_i^j$ denotes the $j$-th neuron/filter at layer $i$. Given a dataset $\mathcal{D} = \{(x_0, y_0), \cdots, (x_n, y_n)\}$ composed of input and output pairs $x_k$ and $y_k$, respectively, the task of training $f_\theta$ is solving the following minimization problem,

$$\min_\theta \frac{1}{n} \sum_{k=1}^{n} E(y_k, f_\theta(x_k)) \tag{1}$$

where $E$ is the error function, $f_\theta(x_k) \in \mathbb{R}^c$ is the softmax final output of $f_\theta$ for a given input $x_k$ and $c$ is the number of classes. A neuron is important if its removal changes the class labels of the input samples compared to the original net. Specifically, the removal of a neuron from the original net may lead to one of the following three main cases.

1. Class score distribution remains the same as in the original network.
2. Class score distribution changes, but the maximum scoring class does not change.
3. Maximum scoring class changes.

For a set of samples, if the removal (exactly one at a time) of three different neurons $N_1, N_2, N_3$ leads to case 1, case 2, and case 3 respectively, then $N_3$ is considered the most important and $N_1$ is the least important neuron. Guided by the above cases, under an i.i.d. assumption, the importance of the $j$-th neuron in the $i$-th layer is determined by,

$$\mathcal{I}_i^j = \sum_{k=1}^{n} L\left(f_\theta(x_k), f_{\theta'}(x_k)\right) \tag{2}$$

$$\text{where, } L = I + |f_\theta(x_k)_q - f_{\theta'}(x_k)_q|$$

$$\text{and, } I = \begin{cases} 1 & \text{if } \arg\max(f_\theta(x_k)) \neq \arg\max(f_{\theta'}(x_k)) \\ 0 & \text{otherwise} \end{cases}$$

Here, $\theta' = (\theta | \theta_i^j = 0)$ denoting that the $j$-th neuron/filter in the $i$-th layer is masked, $d$ is a hyper-parameter, $q = \arg\max(f_\theta(x_k))$ which denotes the index of the class predicted by $f_\theta(x_k)$ and $f_\theta(x_k)_q$ denotes the $q$-th component of the vector $f_\theta(x_k)$. The motivation behind introducing $I$ in the scoring function $L$ is that if masking a neuron/filter causes a misclassification, then it must be considered important for the given task and should be assigned a large importance value. Since $|f_\theta(x_k)_q - f_{\theta'}(x_k)_q| \leq 1$, assigning a value less than 1 will reduce the importance of misclassification while any value greater than 1 has the same effect on the final ranking (the scores might change, but it is rank consistent). Hence, in all our experiments, we assign a value of 1 for misclassifications. On the other hand, if two different neurons/filters produce the same number of misclassifications, as measured by $I$, then the tie is broken by $|f_\theta(x_k)_q - f_{\theta'}(x_k)_q|$ which measures how far the probability of the predicted class deviates before and after masking.

**Advantage of our scoring function over KL divergence:** Measuring the impact of masking a neuron/filter on the final output of a network in order to estimate its importance is not a new concept and in fact, is used by CURL (Luo & Wu, 2020) with a KL-divergence-based criterion. We, instead, define our own criterion as in Eqn. 2. The advantage of our approach over KL-divergence can be understood using the following example. Let us assume that there exists a binary classification task for which a neural network is designed such that the final layer consists of two neurons, representing classes 0 and 1, respectively, with the softmax function being applied to the output. Consider a sample input, $x$ that results in the output vector, $[0.4, 0.6]$, from the pre-trained model denoting that the sample belongs to class 1. Now consider two filters, $p_1$ and $p_2$ wherein masking $p_1$ results in the output vector $[0.6, 0.4]$ and masking $p_2$ results in the output vector $[0.1, 0.9]$. Clearly, $p_1$ is more important than $p_2$ as removing $p_1$ results in a misclassification while removing $p_2$ leads to improved model prediction. We now compute the importance of $p_1$ and $p_2$ for both KL-divergence and our proposed method.

$$\text{KL}\left(f_\theta(x) \parallel f_{\theta|p_1=0}(x)\right) = 0.081 \qquad L\left(f_\theta(x), f_{\theta|p_1=0}(x)\right) = 1.2$$
$$\text{KL}\left(f_\theta(x) \parallel f_{\theta|p_2=0}(x)\right) = 0.310 \qquad L\left(f_\theta(x), f_{\theta|p_2=0}(x)\right) = 0.3$$

As can be seen from the computed scores, using a KL-divergence based criterion instead of our proposed scoring function will produce incorrect rankings of the neurons.

## 3.2 GROUPING MODULE

Computing $\mathcal{I}_i^j$ for individual neurons/filters in a wide and deep network leads to large computational overhead. In order to reduce this overload, we group layerwise similar neurons/filters so that

$\mathcal{I}_i^j$ estimates the importance of the $j$-th group in the $i$-th layer where the size of the group is a hyper-parameter. The similarity between two neurons/filters is measured by the correlation between their output activations/channels. Specifically, let $S_i \in \mathbb{R}^{n \times m}$ be the $i$-th layer's activation output, $n$ the activation values, and $m$ the number of neurons. Then the correlation matrix $C_i \in \mathbb{R}^{m \times m}$ is generated by $C_i = S_i^T S_i$ where the columns of $S_i$ are standardized. The neurons are then partitioned into mutually exclusive groups as follows. We choose a neuron $N$ and the $k$ most similar neurons to $N$ are grouped together to form a single unit, say $G_N$. This process continues until all neurons end up being part of some group. For convolutional neural networks, let $G_i \in \mathbb{R}^{n \times m \times f_w \times f_h}$ denote the output activation channels of the $i$-th layer with $f_w$ and $f_h$ being the width and height of each individual channel. Then $S_i = \sum_{t=1}^{f_w} \sum_{u=1}^{f_h} |G_i^{rstu}|$. This formulation can be generalized to any higher dimensional tensors. Once $S_i$ is computed, the process to form groups is the same as described before.

### 3.3 PRUNING STRATEGIES

Once the neurons/filters are grouped, the importance of each group (from all the layers) is estimated using the ranking module as discussed in Section 3.1. They are then globally sorted across the entire model and the least important ones are pruned until the required number of parameters is achieved. We now present the pruning strategies used in this work for VGG16 (no skip connections), ResNet34, and Bert-base-uncased. The reason for choosing these specific architectures is that most of the vision and language model structures are based on/inspired by these networks. Thus, the strategies used to prune these models can be replicated to prune almost any other network.

**Pruning VGG16:** Pruning networks without skip-connections such as VGG16 is rather straight forward. The ranking module provides a sorted list of groups that need to be pruned. One can simply iterate over the list and discard the least important groups until the user-supplied target parameter budget is reached. The remaining groups form the smaller sub-network.

**Pruning ResNet34:** The ResNet type architectures have two different sets of skip-connections, known as, identity and projection shortcuts (He et al., 2016). Layers with the same number of filters share the identity shortcut while those where a transition between the number of filters is required, for example between the layers having 64 and 128, 128 and 256, 256 and 512 filters, a projection shortcut is introduced. Thus, when iterating over the sorted list provided by the ranking module, if a group from a particular layer is discarded then the least important group from each subsequent layer with an identity shortcut is discarded until a group from a layer after a projection shortcut is encountered.

**Pruning Bert:** In Bert-based architectures, the self-attention layer and the final feed-forward layer in every block share a skip connection. If a group needs to be discarded from either the self-attention layer or the final feed-forward layer, then the least important groups from each of these layers across all blocks in the network need to be discarded. But in doing so, if layer collapse occurs, then the entire network is disposed off. In order to avoid this issue, we first discard groups across the network until a minimum threshold is reached following which the entire block consisting of the current least important group is discarded. Thus, we first prune the network width and then the network depth until the user-supplied target parameter budget is achieved. Here, the threshold for the minimum number of groups is a hyper-parameter.

It should be noted that since we do not set any minimum layer pruning threshold, some of the layers can end up being pruned altogether. This phenomenon is known as layer collapse where entire layers are pruned making a network untrainable (Tanaka et al., 2020). This is an issue, especially for methods that use the preserved weights after pruning for further fine-tuning. Since our sub-networks are re-initialized and trained from scratch (the reason for which is provided in Section 4), layer collapse is not a problem but rather an important aspect of our method. Thus, PvR not only determines the optimal layerwise network width but also the number of layers of the final pruned architecture for a given dataset, pre-trained model, and parameter budget.

## 4 TO RETRAIN OR TO FINE-TUNE?

Once a smaller model is generated, its parameters can be initialized by using either the preserved weights after pruning followed by fine-tuning or re-initialized using well-known random weight

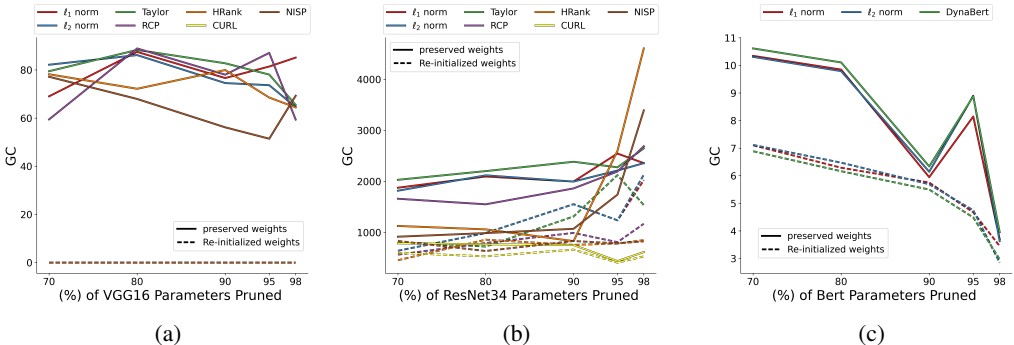

(a)  (b)  (c)

Figure 1: A comparison of the value of GC at different stages of model parameter pruning for (a) VGG16 network on the CIFAR10 dataset (b) ResNet34 network on the TinyImageNet dataset, and (c) Bert base model on the IMDB movie reviews dataset. Fig. (c) is $\log$ transformed.

initialization methods (Glorot & Bengio, 2010; He et al., 2015) followed by re-training also known as training from scratch. Authors Liu et al. (2018) recommend re-initializing and re-training a structurally pruned model as they were able to achieve better test results in most cases using this setup. Instead of using the final model performance as a guideline, we use Geometric Complexity (GC) (Dherin et al., 2022), a recently proposed model complexity measure to determine which initialization works best. GC is defined as,

$$\text{GC} = \frac{1}{n} \sum_{i=1}^{n} \|\nabla_{x_i} f_\theta(x_i)\|_F^2 \qquad (3)$$

where, $x_i$ is input sample data, $n$ is the total number of samples, $f_\theta$ is a model and $f_\theta(x_i)$ is the model output for an input $x_i$. Authors advocate GC to be a proxy for neural network performance and claim that training heuristics such as well-known parameter initialization schemes achieve low GC values. Therefore, to compare the GC of the two separate weight initialization schemes, we first train the VGG16, ResNet34, and Bert-base-uncased models on the CIFAR10, TinyImageNet, and IMDB 50K movie review datasets, respectively, to maximum possible accuracy (see Table 1 for exact accuracy values). The hyper-parameters used for training the models are provided in Section 5.1. Once the base models are trained, $\ell_1$, $\ell_2$ norm, Taylor expansion, HRank, CURL, Random Channel pruning (RCP), and NISP pruning methods are applied to the VGG16 and ResNet34 networks while $\ell_1$ norm, $\ell_2$ norm and DynaBert pruning methods are applied to the Bert-base-uncased network. (Please note that FPGM and LayerDrop do not require any parameter initialization step as they train the original model). Each method prunes the respective models to $70\%, 80\%, 90\%, 95\%,$ and $98\%$ target parameter pruning percentages. After pruning, the models are initialized using the two separate initialization schemes. The resultant GC, right after initialization without any further fine-tuning or re-training, is shown in Fig. 1 which demonstrates that across all models, pruning stages, methods, and datasets, GC for the re-initialization scheme is always lower than its corresponding preserved weights initialization scheme. Thus, the observation made by Liu et al. (2018) that training from scratch after pruning leads to better performance is bolstered by our evidence that re-initialization post-pruning leads to smaller GC. Hence, we choose to re-initialize and train the models recommended by PvR from scratch.

## 5 EXPERIMENTS

### 5.1 SETUP

**Datasets and Models:** All experiments are run on three datasets, namely, CIFAR10 (Krizhevsky et al., 2009), Tiny ImageNet (Le & Yang, 2015) and IMDB 50K movie reviews (Maas et al., 2011). On CIFAR10 we train a VGG16 model (Simonyan & Zisserman, 2014) with batch normalization. On TinyImageNet we train a ResNet34 model (He et al., 2016) while on IMDB 50K movie reviews, we train the Bert-base-uncased model (Devlin et al., 2018). All accuracy scores are reported on the test set of the respective datasets.

**Baselines:** On the CIFAR10 and Tiny ImageNet datasets, we compare our method against $\ell_1$ norm, $\ell_2$ norm, Taylor expansion, HRank, FPGM, CURL, Random Channel pruning (RCP), and NISP. TinyBert is a hand-crafted network having about $90\%$ less parameters than Bert. We train TinyBert from scratch to show how a hand-crafted model fares against a network generated automatically by PvR. LayerDrop stochastically drops layers during training so that the final model is resilient to layer removal during inference. After training, LayerDrop prunes the network depth-wise on demand which leads to the maximum pruning of $70\%$ in the case of Bert.

**Configuration:** We use PyTorch (Paszke et al., 2019) running on an NVIDIA A100 GPU to carry out all the experiments. All three networks are trained to achieve maximum reported accuracy on their respective datasets (results are available in Table 1). For VGG16 and ResNet34, we use SGD with momentum as the optimizer with a momentum value of $0.9$ while for the Bert-base-uncased model, we use the AdamW (Loshchilov & Hutter, 2017) optimizer. The learning rate for each experiment is independently chosen via a grid search over the range $[0.0001, 1.0]$. The cosine annealing scheduler (Loshchilov & Hutter, 2016) is used in tandem with the SGD with momentum optimizer while with AdamW we use the linear warmup scheduler (Goyal et al., 2017). The VGG16 model is trained for 200 epochs with a batch size of 128, the ResNet34 network is trained for 100 epochs with a batch size of 512, and the Bert-base-uncased model is trained for 20 epochs with a batch size of 32. We use the pruning library introduced by Fang et al. (2023) to implement all of the baseline methods. Hyper-parameters specific to each method are taken from the respective works except for RCP where the number of sampled sub-architectures is set to 20 instead of 100 since we do not possess the required resources. We use the same optimizer, scheduler, batch size, and number of epochs for training all methods, including PvR. The learning rate is chosen independently for each experiment via a grid search over the range $[0.0001, 1.0]$. Similar to CURL, we use only a fraction of the entire dataset during the ranking phase of PvR. Specifically, we randomly select 50 training samples per class for both the CIFAR10 and Tiny ImageNet datasets. For the IMDB 50K movie reviews dataset we select 500 samples per class. The group size is set to 2, 4 and 64 for the CIFAR10, Tiny ImageNet, and IMDB 50K movie reviews experiments, respectively. Typically, the group size should be close to 1 with larger values producing faster rankings at the cost of more coarse-grained pruning results. The architecture of each model as recommended by PvR at different pruning percentages is provided in Section A.1 in the Appendix.

## 5.2 RESULTS

Table 1 reports the top-1 accuracy on CIFAR10, top-1 along with top-5 accuracy on Tiny Imagenet, and top-1 accuracy on IMDB 50K movie reviews. The table demonstrates that PvR outperforms all pruning methods across all pruning stages for all three models. Our approach outperforms the VGG16 pre-trained model when $80\%$ and $90\%$ of the model parameters are pruned while achieving less than $1\%$ accuracy drop even after $95\%$ parameter reduction. In fact, surprisingly, our generated Bert-base-uncased sub-networks outperform the original model across multiple pruning stages, especially at the $98\%$ mark. On the CIFAR10 dataset, $\ell_1$ and $\ell_2$ norm pruning approaches turn out to be strong baselines while HRank and CURL give strong competition to PvR. On the TinyImageNet dataset, our pruned ResNet34 networks achieve less than $1\%$ top-5 accuracy drop even after pruning away $90\%$ of the original model size. At the $98\%$ mark, $\ell_1$ norm pruning comes close in terms of top-1 accuracy while $\ell_2$ norm and Taylor expansion pruning approaches come close in terms of top-5 accuracy. On the IMDB 50K movie reviews dataset, the sub-network generated by PvR at the $90\%$ pruning stage outperforms TinyBert, a handcrafted model of similar size.

Additionally, we also report model inference latency which is defined as the amount of time required for a model to make a prediction for a single sample. Inference latency is heavily influenced by the layerwise width of a network as irregular layer structure (not power of two) leads to little improvement in the time required to predict a single sample. We do not report FLOPs count as similarly sized models can have the same FLOPs count but considerably different inference latency (Liu et al., 2021). Tables 2 and 3 report the inference latency, in milliseconds averaged over 1500 runs, of the pruned models generated by each structure pruning method. Pruning techniques that globally rank and remove neurons/filters are more prone to irregular layer widths than their layer-wise pruning counterparts. This is why CURL-based sub-networks are slower than the original unpruned networks. However, PvR-generated sub-networks have substantially reduced inference latency even with irregular layer widths due to reduced depth. This is why in the VGG16 architecture at the $98\%$ pruning mark, the PvR-generated sub-network has significantly low latency while for the ResNet34

Table 1: Comparison of the **Top-1** and **Top-5** accuracy scores on the test set of the CIFAR10, Tiny ImageNet and IMDB 50K movie reviews datasets, for multiple pruning methods at different levels of pruning for the VGG16, ResNet34 and Bert-base-uncased networks. The *model size* row denotes the actual size of the model on disk in megabytes.

| Dataset | Model | Methods | Percentage of Parameters Pruned | | | | | |
|---|---|---|---|---|---|---|---|---|
| | | | 0% | 70% | 80% | 90% | 95% | 98% |
| CIFAR10 | VGG16 (Top-1) | Base | 94.25 | - | - | - | - | - |
| | | $\ell_1$ Norm | - | 93.53 | 92.60 | 92.05 | 90.22 | 87.32 |
| | | $\ell_2$ Norm | - | 93.45 | 92.99 | 91.64 | 90.36 | 87.64 |
| | | Taylor | - | 93.23 | 92.87 | 91.55 | 90.37 | 87.22 |
| | | FPGM | - | 92.72 | 91.64 | 90.85 | 88.41 | 87.09 |
| | | RCP | - | 86.68 | 85.9 | 84.00 | 84.09 | 83.78 |
| | | HRank | - | 92.84 | 93.09 | 92.36 | 91.44 | 91.1 |
| | | CURL | - | 94.18 | 93.89 | 93.49 | 92.14 | 91.66 |
| | | NISP | - | 92.51 | 91.48 | 90.28 | 88.77 | 87.00 |
| | | PvR (ours) | - | **94.21** | **94.33** | **94.32** | **93.64** | **92.18** |
| | | Model Size (MB) | 112 | 33.7 | 22.5 | 11.2 | 5.56 | 2.25 |
| Tiny Imagenet | ResNet34 (Top-1) | Base | 63.02 | - | - | - | - | - |
| | | $\ell_1$ Norm | - | 60.95 | 58.73 | 56.42 | 55.07 | 52.15 |
| | | $\ell_2$ Norm | - | 60.50 | 58.51 | 56.78 | 54.96 | 51.18 |
| | | Taylor | - | 60.65 | 59.4 | 57.06 | 55.16 | 51.59 |
| | | FPGM | - | 60.73 | 59.14 | 56.22 | 52.88 | 50.42 |
| | | RCP | - | 58.18 | 56.52 | 54.21 | 50.65 | 45.46 |
| | | HRank | - | 57.90 | 55.30 | 52.85 | 50.55 | 46.83 |
| | | CURL | - | 58.38 | 56.85 | 55.13 | 48.33 | 39.49 |
| | | NISP | - | 57.06 | 54.76 | 52.62 | 50.82 | 45.10 |
| | | PvR (ours) | - | **62.68** | **61.47** | **59.54** | **58.27** | **52.49** |
| Tiny Imagenet | ResNet34 (Top-5) | Base | 83.22 | - | - | - | - | - |
| | | $\ell_1$ Norm | - | 81.59 | 80.65 | 79.30 | 79.16 | 76.68 |
| | | $\ell_2$ Norm | - | 81.28 | 80.58 | 79.40 | 78.32 | 77.42 |
| | | Taylor | - | 81.68 | 80.61 | 79.79 | 78.86 | 77.42 |
| | | FPGM | - | 81.68 | 80.39 | 78.70 | 76.46 | 76.13 |
| | | RCP | - | 80.51 | 79.26 | 78.08 | 75.57 | 72.19 |
| | | HRank | - | 80.27 | 78.85 | 77.02 | 75.68 | 73.26 |
| | | CURL | - | 80.68 | 80.22 | 79.81 | 74.80 | 67.46 |
| | | NISP | - | 79.39 | 77.56 | 76.78 | 75.35 | 71.89 |
| | | PvR (ours) | - | **83.08** | **82.84** | **82.34** | **81.59** | **77.97** |
| | | Model Size (MB) | 163 | 46.8 | 30.6 | 15.2 | 8.03 | 2.72 |
| IMDB 50K movie reviews | Bert-base-uncased (Top-1) | Base | 82.24 | - | - | - | - | - |
| | | TinyBert | - | - | - | 82.48 | - | - |
| | | $\ell_1$ Norm | - | 82.16 | 82.18 | 81.86 | 82.00 | 81.57 |
| | | $\ell_2$ Norm | - | 82.14 | 81.97 | 81.93 | 82.18 | 81.90 |
| | | LayerDrop | 80.74 | 80.61 | - | - | - | - |
| | | DynaBert | - | 81.78 | 81.96 | 81.42 | 80.89 | 80.00 |
| | | PvR (ours) | - | **82.41** | **82.02** | **82.60** | **82.10** | **82.56** |
| | | Model Size (MB) | 1220 | 339 | 246 | 130 | 73.4 | 36.1 |

architecture, our recommended sub-networks have much lower inference latency than all other methods across all pruning stages. Experiments with the inference latency of Bert-base-uncased pruned networks, as shown in Table 3, reveal that PvR offers the best accuracy versus inference latency trade-off.

We perform an ablation study to observe the change in accuracy of pruned networks with increasing group size as well as the time required for pruning under varying group sizes. Specifically, we look at the $98\%$ pruned version of the VGG16 and ResNet34 networks for group sizes varying from $2-32$ and $4-32$, respectively. We perform this ablation study for only the maximum level of pruning as it is the worst-case scenario in terms of both accuracy and time. According to Fig. 2a and 3a in Section A.2 of the Appendix, PvR is generally robust to group size as the top-1 accuracy drop from group size of 2 to 32 is about $1\%$ and the top-5 accuracy drop from group size of 4 to 32 is about $2\%$. At the same time, with a group size of 32, PvR is able to prune VGG16 and ResNet34 under 20 seconds and 12 minutes, respectively, as demonstrated in Fig. 2b and 3b.

Table 2: Comparison of the inference latency of pruned sub-networks generated by multiple pruning techniques for the VGG16 and ResNet34 models. The results are reported in terms of milliseconds averaged over 1500 runs with a standard deviation of $0.01-0.001$. Smaller values are better.

| Model | Pruning Level | Base | $\ell_1$ | $\ell_2$ | Taylor | FPGM | RCP | HRank | CURL | NISP | PvR |
|---|---|---|---|---|---|---|---|---|---|---|---|
| VGG16 | 0% | 1.72 | - | - | - | - | - | - | - | - | - |
|  | 70% | - | 1.78 | 1.78 | 1.78 | 1.71 | **1.59** | 1.78 | 1.75 | 1.78 | 1.68 |
|  | 80% | - | 1.82 | 1.82 | 1.82 | 1.70 | **1.55** | 1.82 | 1.74 | 1.82 | 1.69 |
|  | 90% | - | 1.76 | 1.76 | 1.76 | 1.72 | 1.53 | 1.76 | 1.88 | 1.76 | **1.13** |
|  | 95% | - | 1.68 | 1.68 | 1.68 | 1.68 | 1.52 | 1.68 | 1.84 | 1.68 | **1.04** |
|  | 98% | - | 1.47 | 1.47 | 1.47 | 1.55 | 1.49 | 1.47 | 1.69 | 1.47 | **0.95** |
| ResNet34 | 0% | 4.03 | - | - | - | - | - | - | - | - | - |
|  | 70% | - | 4.20 | 4.20 | 4.20 | 5.097 | 4.24 | 4.20 | 5.83 | 4.20 | **2.69** |
|  | 80% | - | 4.13 | 4.13 | 4.13 | 4.99 | 4.16 | 4.13 | 5.75 | 4.13 | **2.59** |
|  | 90% | - | 3.95 | 3.95 | 3.95 | 4.98 | 4.10 | 3.95 | 5.91 | 3.95 | **2.60** |
|  | 95% | - | 3.89 | 3.89 | 3.89 | 4.88 | 4.02 | 3.89 | 5.88 | 3.89 | **2.60** |
|  | 98% | - | 3.39 | 3.39 | 3.39 | 4.24 | 3.53 | 3.39 | 4.67 | 3.39 | **2.54** |

Table 3: A comparison of the inference latency of pruned sub-networks generated by multiple pruning techniques for the Bert-base-uncased model. The results are reported in terms of milliseconds averaged over 1500 runs with a standard deviation range of $0.01-0.001$. Smaller values are better.

| Model | Pruning Level | Base | $\ell_1$ | $\ell_2$ | TinyBert | LayerDrop | DynaBert | PvR |
|---|---|---|---|---|---|---|---|---|
| Bert-base-uncased | 0% | 5.63 | - | - | - | - | - | - |
|  | 70% | - | 5.37 | 5.37 | - | 5.74 | 5.39 | **5.25** |
|  | 80% | - | 5.37 | 5.37 | - | - | 5.43 | **5.14** |
|  | 90% | - | 4.98 | 4.98 | **2.01** | - | 1.58 | 4.03 |
|  | 95% | - | 4.98 | 4.98 | - | - | **0.70** | 1.58 |
|  | 98% | - | 4.89 | 4.89 | - | - | **0.67** | 1.50 |

## 6 CONCLUSION

In this work, we propose Pruning via Ranking (PvR) a novel, global structured pruning approach that reduces both parameters as well as depth and generates dense sub-networks that comply with any user-supplied parameter budget. PvR consists of a grouping module and a ranking module that are used to efficiently estimate the importance of groups of neurons/filters. The pruning strategies for the VGGG16, ResNet34, and Bert-base-uncased models are also shown. We use a recently proposed model complexity measure, Geometric Complexity (GC) to show that re-initialization after pruning is a better training heuristic. Finally, the pruned networks are re-initialized and trained from scratch. Through exhaustive comparisons, in terms of accuracy and model inference latency, against diverse pruning approaches on benchmark datasets with standard models, we demonstrate the efficacy of our approach. We also show that PvR is robust to group size.

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

# A APPENDIX

## A.1 PRUNED ARCHITECTURES

### A.1.1 VGG16

Table 4: The number of channels per layer for each pruning percentage. Here, *"M"* denotes the position of the max-pooling layer.

| Pruned Amount | Architecture |
|---|---|
| 0% | 64, 64, "M", 128, 128, "M", 256, 256, 256, "M", 512, 512, 512, "M", 512, 512, 512, "M" |
| 70% | 58, 64, "M", 126, 128, "M", 238, 224, "M", 192, 192, 94, "M", 56, 482, 512, "M" |
| 80% | 58, 64, "M", 126, 128, "M", 238, 224, "M", 192, 192, 94, "M", 56, 196, 512, "M" |
| 90% | 54, 64, "M", 124, 128, "M", 224, 220, "M", 174, 110, "M" |
| 95% | 42, 64, "M", 110, 126, "M", 170, 138, 92, "M" |
| 98% | 36, 62, "M", 78, 112, "M", 92, 44, 46, "M" |

### A.1.2 RESNET34

Table 5: The number of channels per layer per block with the number of blocks being denoted by $\times$ and each block being denoted by *[.]*.

| Pruned Amount | Architecture |
|---|---|
| 0% | [64, 64]$\times$3, [128, 128]$\times$4, [256, 256]$\times$6, [512, 512]$\times$3 |
| 70% | [60, 60]$\times$3, [112, 112]$\times$4, [328, 328]$\times$3 |
| 80% | [60, 60]$\times$3, [104, 104]$\times$4, [256, 256]$\times$3 |
| 90% | [60, 60]$\times$3, [100, 100]$\times$4, [152, 152]$\times$3 |
| 95% | [60, 60]$\times$3, [76, 76]$\times$4, [88, 88]$\times$3 |
| 98% | [60, 60]$\times$3, [36, 36]$\times$4, [44, 44]$\times$3 |

### A.1.3 BERT-BASE-UNCASED

Table 6: The first value in the *architecture* column indicates the hidden size, the second value (or list of values) indicates the intermediate size (or the blockwise intermediate sizes), the third value is the number of attention heads and the final value is the number of hidden layers. These values are provided in the form of the HuggingFace config.json file

| Pruned Amount | Architecture |
|---|---|
| 0% | 768, 3072, 12, 12 |
| 70% | 256, [2944, 3008, 3008, 2944, 3072, 2944, 3008, 2944, 3072, 3008, 3008, 2944], 4, 12 |
| 80% | 192, [2944, 3008, 3008, 2944, 3072, 2944, 3008, 2944, 3072, 2880, 3008, 2944], 3, 12 |
| 90% | 128, [2944, 3008, 2944, 3008, 2944, 3008, 2560, 2944, 2944], 2, 9 |
| 95% | 128, [2880, 2880, 2880], 2, 3 |
| 98% | 64, [2880, 2880, 2880], 2, 3 |

## A.2 ABLATION STUDY

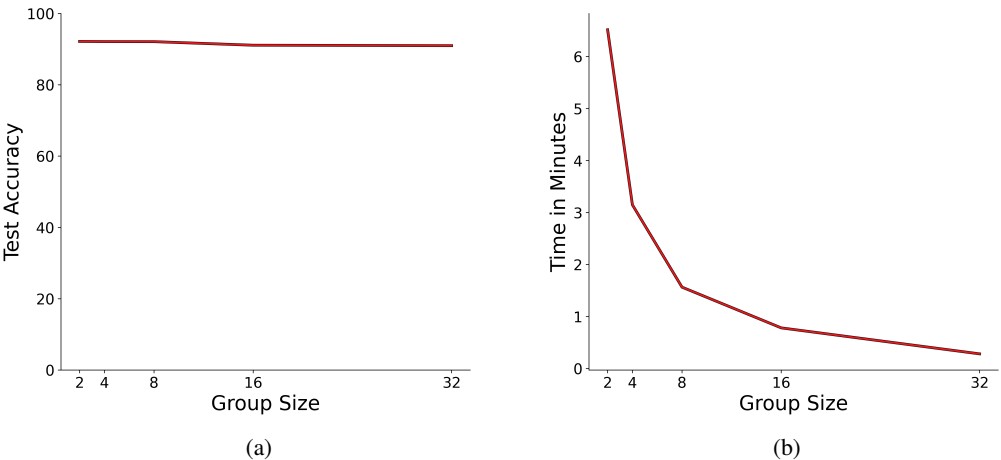

(a)                                        (b)

Figure 2: (a) Change in Top-1 accuracy of the $98\%$ pruned VGG16 network with increasing group size trained on CIFAR10 (b) The time required by PvR to prune $98\%$ of the parameters of the VGG16 network on the CIFAR10 dataset with increasing group size

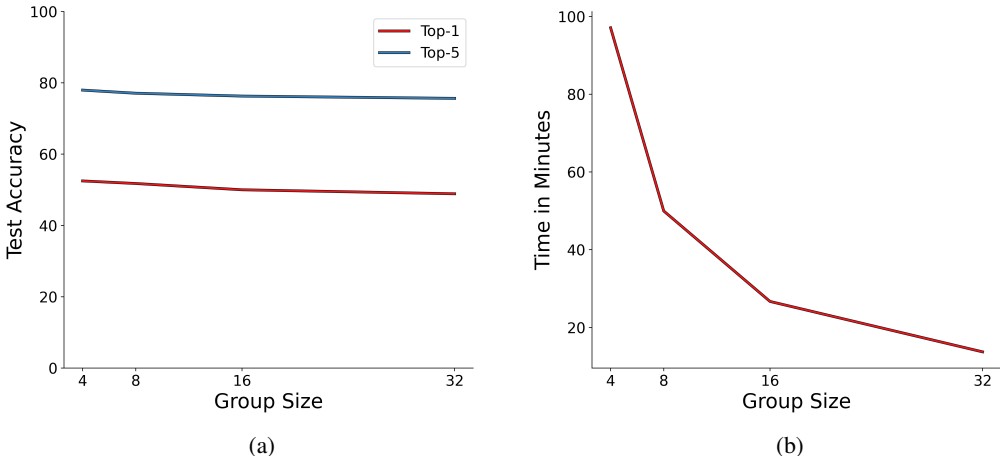

(a)                (b)

Figure 3: (a) Change in Top-1 and Top-5 accuracy of the $98\%$ pruned ResNet34 network with increasing group size trained on TinyImageNet (b) The time required by PvR to prune $98\%$ of the parameters of the ResNet34 network on the TinyImageNet dataset with increasing group size

