# OpenReview forum: "Pruning via Ranking (PvR): A unified structured pruning approach"
_ICLR.cc/2024/Conference — ICLR 2024 Conference Withdrawn Submission_

### Official Review · Reviewer_r6fS · 2023-10-31

**Soundness:** 3 good
**Presentation:** 3 good
**Contribution:** 2 fair
**Rating:** 6
**Confidence:** 3

**Summary:**

This paper presents a structured pruning approach named Pruning via Ranking (PvR) that can prune the models according to user-assigned budget. PvR has one grouping module to group neurons for speeding up pruning process and one ranking module to estimate the importance of grouped neurons. It can reduce model depth, leading to training and inference efficiency wins. Apart from this, the authors suggest to retrain the subnet from scratch and proposes a metric called Geometric Complexity to measure its efficacy. The authors validated the PvR on several classic models comparing with other pruning methods.

**Strengths:**

1. PvR can work for both NLP and vision models
2. PvR can prune with user-provided parameter budget.

**Weaknesses:**

Lacks validation on the proposed grouping module. How can different grouping settings impact the performance of pruned model? How much it can speed up the overall process?

**Questions:**

1. What makes PvR perform well on both vision and NLP models? Why couldn't other pruning methods achieve this? Would like to hear more insights on this claim.
2. How long does Grouping and Ranking processes each take? How does the total time PvR takes compared to other pruning methods?
3. Is there any strategy to initialize and retrain the pruned model (does it have any relationship, like some proportional to the original model)?

---

> ### Author Response · Authors · 2023-11-15
>
> We thank the reviewer for taking the time to review our work.
>
> Q. Lacks validation on the proposed grouping module. How can different grouping settings impact the performance of pruned model? How much it can speed up the overall process?
>
> A. Based on the feedback received, we have added an ablation study to observe the change in accuracy of pruned networks with increasing group size as well as the time required for pruning under varying group sizes. Specifically, we look at the $98$% pruned version of the VGG16 and ResNet34 networks for group sizes varying from $2-32$ and $4-32$, respectively. We perform this ablation study for only the maximum level of pruning as it is the worst-case scenario in terms of both accuracy and time. According to Fig. 2a and 3a in Section A.2 of the Appendix, PvR is generally robust to group size as the top-1 accuracy drop from group size of $2$ to $32$ is about $1$% and the top-5 accuracy drop from group size of $4$ to $32$ is about $2$%. At the same time, with a group size of $32$, PvR is able to prune VGG16 and ResNet34 under $20$ seconds and $12$ minutes, respectively, as demonstrated in Fig. 2b and 3b.
>
> Q. What makes PvR perform well on both vision and NLP models? Why couldn't other pruning methods achieve this? Would like to hear more insights on this claim.
>
> A. We are delighted by this insightful question posed by the reviewer. The main idea of pruning is to generate a smaller model that performs at par with the unpruned network. The brute-force solution to this task is to mask each neuron/filter/weight at a time and look at the change in the final output. If the neuron/filter/weight causes a large change in the output then it can marked as important. Most of the works in literature avoid solving the pruning problem in this way as it is extremely inefficient and requires multiple forward passes over the entire data to compute the importance for a single neuron/filter/weight. PvR takes this exact brute-force route but avoids the large computational overhead by requiring a very small subset of the data to determine the importance of neurons/filters/weights. It also uses a carefully designed grouping module to mask multiple neurons/filters at the same time. CURL$^{[1]}$ is another method which solves the problem in a similar way but we have shown in our work that the ranking function proposed by us is superior as it takes all possible scenarios into account. We urge the reviewer to look at the ablation study that has been added to the manuscript wherein we demonstrate the robustness of PvR to group size as well as the small computation time required to prune large networks.
>
> Q. How long does Grouping and Ranking processes each take? How does the total time PvR takes compared to other pruning methods?
>
> A. We urge the reviewer to look at the ablation study that has been added to the manuscript wherein we demonstrate the wall-clock time required to prune large networks using PvR.
>
> Q. Is there any strategy to initialize and retrain the pruned model (does it have any relationship, like some proportional to the original model)?
>
> A. The pruned networks are initialized using the same process as the unpruned models where the weight initialization strategy for each unpruned model is reported in their respective papers. We use the latest weight initialization techniques recommended by PyTorch. The recipe for training both pruned and unpruned models is exactly the same. These details are provided in the Configuration sub-section under Section 5.1.

---

### Official Review · Reviewer_Huai · 2023-11-08

**Soundness:** 2 fair
**Presentation:** 2 fair
**Contribution:** 2 fair
**Rating:** 3
**Confidence:** 4

**Summary:**

This paper introduces a new pruning criterion based ranking. The ranking is achieved by a carefully designed importance function that pays more attention to those parameters that change the prediction of the network. An additional term  is introduced to penalise prediction changes. The experiments show positive results on CIFAR-10 and TinyImageNet.

**Strengths:**

This approach introduces an interesting ranking function. It would be beneficial if the author could offer a more detailed analysis of the ranking outcomes. Specifically, it would be intriguing to explore the ramifications of eliminating weights that do not result in incorrect predictions. What impact would this have on the overall model performance and efficiency?

**Weaknesses:**

The paper conduct empirical experiments on CIFAR-10 and Tiny-ImageNet, both of which are relatively small-scale datasets. It raises concerns about the generalizability and robustness of the proposed method.  It would be advantageous if the author could present additional results using the ImageNet-1K dataset, which is more complex and varied. Furthermore, the paper currently lacks a thorough analysis of the proposed ranking method, which is crucial for understanding the efficacy and underlying mechanics of the technique. Including such an analysis would significantly enhance the paper's contribution to the field.

**Questions:**

Please see weakness above.

---

> ### Author Response · Authors · 2023-11-15
>
> We thank the reviewer for taking the time out to review our work.
>
> Q. Specifically, it would be intriguing to explore the ramifications of eliminating weights that do not result in incorrect predictions. What impact would this have on the overall model performance and efficiency?
>
> A. We would like to bring to the reviewer’s notice that PvR currently removes those neurons/filters that do not cause any change in the final output, i.e., we remove neurons/filters that do not result in incorrect predictions. The entire objective of this work is to achieve this goal.
>
> Q. The paper conduct empirical experiments on CIFAR-10 and Tiny-ImageNet, both of which are relatively small-scale datasets. It raises concerns about the generalizability and robustness of the proposed method. It would be advantageous if the author could present additional results using the ImageNet-1K dataset, which is more complex and varied.
>
> A. Unfortunately, we do not have the resources to run all the methods described in the paper along with PvR on the ImageNet-1K dataset within the rebuttal time frame. This is the reason why we demonstrated the efficacy of our approach on three different datasets and models including both vision as well as NLP. At the same time, we would like to bring to the reviewer’s notice that multiple pruning works have been published in the ICLR conference by producing results on CIFAR10 and/or CIFAR100 and TinyImageNet. We provide some references for the same.
>
> [1] REVISITING PRUNING AT INITIALIZATION THROUGH THE LENS OF RAMANUJAN GRAPH (2022)
>
> [2] OVER-PARAMETERIZED MODEL OPTIMIZATION WITH POLYAK-ŁOJASIEWICZ CONDITION (2023)
>
> [3] DEPTHFL: DEPTHWISE FEDERATED LEARNING FOR HETEROGENEOUS CLIENTS (2023)
>
> [4] PRUNING DEEP NEURAL NETWORKS FROM A SPARSITY PERSPECTIVE (2023)
>
> [5] SPARSITY WINNING TWICE: BETTER ROBUST GENERALIZATION FROM MORE EFFICIENT TRAINING (2022)
>
> [6] SIGNING THE SUPERMASK: KEEP, HIDE, INVERT (2022)
>
> Q. Furthermore, the paper currently lacks a thorough analysis of the proposed ranking method, which is crucial for understanding the efficacy and underlying mechanics of the technique. Including such an analysis would significantly enhance the paper's contribution to the field.
>
> A. Based on the feedback received, we have added an ablation study to observe the change in accuracy of pruned networks with increasing group size as well as the time required for pruning under varying group sizes. Specifically, we look at the $98$% pruned version of the VGG16 and ResNet34 networks for group sizes varying from $2-32$ and $4-32$, respectively. We perform this ablation study for only the maximum level of pruning as it is the worst-case scenario in terms of both accuracy and time. According to Fig. 2a and 3a in Section A.2 of the Appendix, PvR is generally robust to group size as the top-1 accuracy drop from group size of $2$ to $32$ is about $1$% and the top-5 accuracy drop from group size of $4$ to $32$ is about $2$%. At the same time, with a group size of $32$, PvR is able to prune VGG16 and ResNet34 under $20$ seconds and $12$ minutes, respectively, as demonstrated in Fig. 2b and 3b. We would also like to bring to the reviewer's notice that $d$ as a hyper-parameter in Eqn. 2 is misleading as its value has to always be set to $1$ which is what we have done for all our experiments. Hence we have updated the manuscript by replacing $d$ with a constant value of $1$ in Eqn. $2$. We also request the reviewer to provide us with more details on any specific analysis they are looking for which could further improve the quality of our work.

---

### Official Review · Reviewer_gzCT · 2023-11-08

**Soundness:** 2 fair
**Presentation:** 4 excellent
**Contribution:** 2 fair
**Rating:** 3
**Confidence:** 5

**Summary:**

The paper proposes a new pruning method which can prune neurons, channels and layers. The authors claim that the method apply to both vision (VGG and ResNet) and NLP models (Bert). The core steps of the proposed pruning are:
1. Evaluate the importance of each neuron/channel/layer, etc. The importance metric is shown in Eq.2 of the original paper. It consists of two parts: (1) The output's L1 error before and after pruning a certain neuron (rescontruction error). (2) A constant error (hyper-parameter) if pruning a certain neuron leads to a different prediction result.
2. Group similar neurons/channels according to the correlation.
3. Prune unimportant groups. While the authors do not explain how to compute group importance according to neuron importance. I just assume the most naive "average" method is used.
Experiments are done on CIFAR-10, TinyImageNet, and IMDB.

**Strengths:**

1. The paper is well-written and easy to understand.
2. The authors propose to takes that "whether the prediction result changes" as a parts of importance metric. While most previous papers only concider the change of the output.
3. Experiments are done on both vision and NLP tasks.

**Weaknesses:**

I think the paper has some obvious flaws:
1. Lack of essential ablation studies. The authors introduce two very important hyperparameters: the constant error "d" and the group size. I think they will have a great influence on the final accuracy. The authors claim their algorithm applys to both vision and NLP models. However, it would meaningless if we have to carefully adjust these two hyper-parameters for the best accuracy on different models and tasks. This is my main concern.
2. The authors claim the proposed algorithms consider the paramete budget. While as I know, all existing pruning methods can achieve this because it is a basic requirement. Take L1 norm based pruing as an example, you can consider the parameter budget by adjust the pruning threshold (This process is very fast with ignorable cost).
3. Limited creativity. GC based initialization method is proposed by another paper, this paper only uses it.
4. VGG-16 is a very redundant model for CIFAR-10 dataset. The has been a consensus in the field. I doubt whether the corresponding results are still convicing.
5. I am not sure but the authors may not describe how to compute group importance according to neuron/channel/.. importance.
6. Lack of some details. As the paper says, the proposed method achieves lower latency because of layer pruning. Could you please describe in which case the proposed methed will prune an entire layer?
7. Lack of reference papers such as [1] (correlation based pruning) and [2] (reconstruction error based pruning).
[1] COP: Customized Deep Model Compression via Regularized Correlation-Based Filter-Level Pruning. IJCAI 2019
[2] ThiNet: A Filter Level Pruning Method for Deep Neural Network Compression

**Questions:**

Please see the weaknesses for details.

---

> ### Author Response · Authors · 2023-11-15
>
> We would like to thank the reviewer for taking the time to review our work.
>
> Q. Lack of essential ablation studies. The authors introduce two very important hyperparameters: the constant error "d" and the group size. I think they will have a great influence on the final accuracy. The authors claim their algorithm applys to both vision and NLP models. However, it would meaningless if we have to carefully adjust these two hyper-parameters for the best accuracy on different models and tasks. This is my main concern.
>
> A. We thank the reviewer for bringing this to light. As explained in Section 3.2 after Eqn. $2$, the value of $d$ needs to be set to $1$ irrespective of the model and dataset. In fact, this is the value that we chose for all our experiments. The reasoning for this exact value is also present in the manuscript in Section 3.2. We agree that calling $d$ as a hyper-parameter is misleading and have updated the manuscript by replacing $d$ with a constant value of $1$ in Eqn. $2$.
> As explained in the configuration subsection of section 5.2, ideally you would want to set the group size close to 1 while higher values produce faster rankings at the cost of more coarse-grained results. This is a hyper-parameter that needs to be set according to the user’s time and hardware budget. We have updated the manuscript with an ablation study to observe the change in accuracy of pruned networks with increasing group size as well as the time required for pruning under varying group sizes. Specifically, we look at the $98$% pruned version of the VGG16 and ResNet34 networks for group sizes varying from $2-32$ and $4-32$, respectively. We perform this ablation study for only the maximum level of pruning as it is the worst-case scenario in terms of both accuracy and time. According to Fig. 2a and 3a in Section A.2 of the Appendix, PvR is generally robust to group size as the top-1 accuracy drop from group size of $2$ to $32$ is about $1$% and the top-5 accuracy drop from group size of $4$ to $32$ is about $2$%. At the same time, with a group size of $32$, PvR is able to prune VGG16 and ResNet34 under $20$ seconds and $12$ minutes, respectively, as demonstrated in Fig. 2b and 3b.
>
> Q. The authors claim the proposed algorithms consider the paramete budget. While as I know, all existing pruning methods can achieve this because it is a basic requirement. Take L1 norm based pruing as an example, you can consider the parameter budget by adjust the pruning threshold (This process is very fast with ignorable cost).
>
> A. We respectfully disagree with the reviewer. A number of algorithms such as L1 norm, as you rightly pointed out, have a parameter that needs to be tuned by performing multiple forward passes in order to achieve the precise parameter budget before beginning the pruning process. For techniques like L1 norm this process is fast while for methods such as HRank which finds the rank of the activations, it is slow. Other algorithms such as LayerDrop are unable to achieve all possible parameter budgets while methods like OTO$^{[1]}$ provide no such parameter. Our algorithm, on the other hand, explicitly considers a given user-specified parameter budget to prune the larger network.
>
> Q. Limited creativity. GC based initialization method is proposed by another paper, this paper only uses it.
>
> A. We have not claimed that GC is a measure proposed by us. The paper titled "Rethinking the value of network pruning"$^{[2]}$ empirically demonstrated that re-initializing a structurally pruned network and training from scratch is a better heuristic than fine-tuning the preserved weights. We used GC, a recently proposed model complexity measure, to shed light on this. Based on the fact that lower GC is a proxy for better performance in deep learning, we made the informed decision to re-initialize and train from scratch instead of simply relying on the performance of a model.
>
>
> [1] Chen, Tianyi, Bo Ji, Tianyu Ding, Biyi Fang, Guanyi Wang, Zhihui Zhu, Luming Liang, Yixin Shi, Sheng Yi, and Xiao Tu. "Only train once: A one-shot neural network training and pruning framework." Advances in Neural Information Processing Systems 34 (2021): 19637-19651.
>
> [2] Liu, Zhuang, Mingjie Sun, Tinghui Zhou, Gao Huang, and Trevor Darrell. "Rethinking the value of network pruning." arXiv preprint arXiv:1810.05270 (2018).

---

> > ### Author Response · Authors · 2023-11-15
> >
> > Q. VGG-16 is a very redundant model for CIFAR-10 dataset. The has been a consensus in the field. I doubt whether the corresponding results are still convicing.
> >
> > A. We respectfully but strongly disagree with the reviwer since the following recent pruning papers that were published at ICLR apply VGG16 on CIFAR10,
> >
> > [1] REVISITING PRUNING AT INITIALIZATION THROUGH THE LENS OF RAMANUJAN GRAPH (2022)
> >
> > [2] CRAM: A COMPRESSION-AWARE MINIMIZER (2023)
> >
> > [3] DFPC: Data flow driven pruning of coupled channels without data (2023)
> >
> > [4] OTOV2: AUTOMATIC, GENERIC, USER-FRIENDLY (2023)
> >
> > [5] OVER-PARAMETERIZED MODEL OPTIMIZATION WITH POLYAK-ŁOJASIEWICZ CONDITION (2023)
> >
> > We would be really thankful to the reviewer if they could provide a reference to said consensus on VGG16 as we are unaware of any such consensus.

---

> > ### Comment · Reviewer_gzCT · 2023-11-23
> >
> > 1. “the value of d needs to be set to 1 irrespective of the model and dataset.”
> > I have noticed the claim. Thanks for the authors' response.
> >
> > 2. "We perform this ablation study for only the maximum level of pruning as it is the worst-case scenario in terms of both accuracy and time"
> > Thanks for the authros response. In Figure 2(a) and Figure 3(a) of the appendix, I think the range of of y-axis should be limited around 92% or so to clearly show the accuracy's change. Besides, the group size is 4 for TinyImageNet datasets as described in the paper, which will take more than 100 minutes to confirm the pruned elements.
> >
> > 3. "For techniques like L1 norm this process is fast while for methods such as HRank which finds the rank of the activations, it is slow".
> > As shown in Table 1 of the original paper, all pruning methods are pruned with the same "Percentage of Parameters Pruned". As such, at least all these pruning algorithms can cutomize the pruning ratio. I recommend the authors to compare the pruning time with them. (Time for confirming the pruned elements, because I think the fine-tuning time should be the same)
> >
> > 4. "We have not claimed that GC is a measure proposed by us".
> > Sorry for ambiguous comments. I acknowledge that the authors have not claimed that GC is their proposed measure. What I mean is, considering GC is not one of the contribution of the paper, the creativity of the paper (other parts in the paper) is limited.

---

> ### Author Response · Authors · 2023-11-15
>
> Q. I am not sure but the authors may not describe how to compute group importance according to neuron/channel/.. importance.
>
> A. The process of computing the importance of a single neuron is explained in Section 3.1. The process of computing groups is explained in Section 3.2. Then, the first line in Section 3.3 explains how the ranking of each group is computed. Just like we mask a single neuron to compute its importance, in the same way, we mask all neurons in a group (at the same time) and then compute the importance of that group. There is no "averaging" involved.
>
> Q. Lack of some details. As the paper says, the proposed method achieves lower latency because of layer pruning. Could you please describe in which case the proposed methed will prune an entire layer?
>
> A. The first few lines in Section 3.3 explain how the groups are removed based on the ranking, i.e., once the groups are formed and ranked, they are globally sorted and the least important groups are removed. In the last paragraph, of the same section we then explain that removing groups after a global sort can lead to the removal of all neurons/filters from a layer rendering that layer “collapsed” since we do not set any minimum layer pruning threshold. The removal of a layer entirely depends on the pruning budget, the model, and the dataset. Our algorithm PvR automatically decides and removes specific layers as and when required.
>
> Q. Lack of reference papers such as [1] (correlation based pruning) and [2] (reconstruction error based pruning). [1] COP: Customized Deep Model Compression via Regularized Correlation-Based Filter-Level Pruning. IJCAI 2019 [2] ThiNet: A Filter Level Pruning Method for Deep Neural Network Compression
>
> A. We would like to bring to the reviewer's notice that the paper ThiNet has already been cited in the manuscript. We already have 20+ citations in our work but we can add three more if the reviewer feels that citing these papers can improve the quality of our manuscript.

---

### Official Review · Reviewer_L5F6 · 2023-11-12

**Soundness:** 2 fair
**Presentation:** 2 fair
**Contribution:** 2 fair
**Rating:** 3
**Confidence:** 3

**Summary:**

This paper presents a neural network pruning method. The importance of the network parameters is sorted based on their importance to the prediction result change before and after pruning. The model parameters are grouped to reduce computational cost. Pruning strategies are discussed for VGG 16, ResNet 34, and Bert model.

**Strengths:**

A novel network pruning method is proposed and verified by experiments.

**Weaknesses:**

1. The algorithm introduction is hard to follow. The missing of algorithm flowchart makes the reviewer difficult to have a clear idea about how exactly the described components work together.
2. Even though grouping is used to reduce computational cost, the cost of computational cost can still be quite high. There is no empirical analysis of time cost in the experiment.
3. The baseline models compared in the experiment are not quite state-of-the-art. More recent models need to be covered. Some of the existing methods are relevant to this work, such as
Kuang J, Shao M, Wang R, et al. Network pruning via probing the importance of filters[J]. International Journal of Machine Learning and Cybernetics, 2022, 13(9): 2403-2414.

**Questions:**

Please refer to the comments.

---

> ### Author Response · Authors · 2023-11-15
>
> We would like to thank the reviewer for taking the time to review our work.
>
> Q. The algorithm introduction is hard to follow. The missing of algorithm flowchart makes the reviewer difficult to have a clear idea about how exactly the described components work together.
>
> A. We respectfully disagree with the reviewer that a flowchart is required for this work. PvR consists of only two modules, a grouping and a ranking module. The grouping module is responsible for grouping similar neurons/filters into a single group following which the ranking module sets an importance value on each of those groups. Based on the importance scores, the groups are then sorted and the least important ones are removed. This is already explained in the fourth paragraph in Section 1, as well as in the first few lines in Section 3.3.
>
> Q. Even though grouping is used to reduce computational cost, the cost of computational cost can still be quite high. There is no empirical analysis of time cost in the experiment.
>
> A. We thank the reviewer for highlighting this. We have updated the manuscript with an ablation study on the time required for pruning with varying group sizes. Specifically, we look at the time required by PvR for pruning the VGG16 and ResNet34 networks at the maximum pruning level under group sizes of $2,4,8,16,32$ and $4,8,16,32$, respectively. We find that PvR can prune VGG16 in under $20$ seconds with a group size of $32$ with only $1$% drop in performance in comparison to the top-1 accuracy of the network pruned with a group size of $2$. Similarly, PvR can prune ResNet34 in under $12$ minutes with a group size of $32$ with about $2$% drop in performance in comparison to the top-5 accuracy of the network pruned with a group size of $4$. Please look at Figs. 2(b) and 3(b) in Appendix.
>
> Q. The baseline models compared in the experiment are not quite state-of-the-art. More recent models need to be covered. Some of the existing methods are relevant to this work, such as Kuang J, Shao M, Wang R, et al. Network pruning via probing the importance of filters[J]. International Journal of Machine Learning and Cybernetics, 2022, 13(9): 2403-2414.
>
> A. We compare our method against $11$ different pruning techniques from 2016 to 2022 with Random Channel Pruning (RCP) being the latest (Li, Yawei, Kamil Adamczewski, Wen Li, Shuhang Gu, Radu Timofte, and Luc Van Gool. "Revisiting random channel pruning for neural network compression." In Proceedings of the IEEE/CVF Conference on Computer Vision and Pattern Recognition, pp. 191-201. 2022.). We believe that RCP is a stronger and better-known baseline in comparison to the one provided by the reviewer.

---

### Author Response · Authors · 2023-11-17

We request the reviewers to please respond to our rebuttal and provide us with a fair chance at the reviewing process.

---

### Author Response · Authors · 2023-11-18
**Please respond to our rebuttal**

We request the reviewers to please respond to our rebuttal at the very least.

---

### Meta-Review · Area_Chair_35q3 · 2023-12-14

**Metareview:**

The paper proposes a new pruning method which can prune neurons, channels and layers. The authors claim that the method apply to both vision (VGG and ResNet) and NLP models (Bert). The core steps of the proposed pruning are:
1. Evaluate the importance of each neuron/channel/layer, etc. The importance metric is shown in Eq.2 of the original paper. It consists of two parts: (1) The output's L1 error before and after pruning a certain neuron (rescontruction error). (2) A constant error (hyper-parameter) if pruning a certain neuron leads to a different prediction result.
2. Group similar neurons/channels according to the correlation.
3. Prune unimportant groups. While the authors do not explain how to compute group importance according to neuron importance. I just assume the most naive "average" method is used. Experiments are done on CIFAR-10, TinyImageNet, and IMDB.

Strengths：
1. I think the paper is technically written for most parts.
2. Experiments are done on both vision and NLP tasks.

Weaknesses：
1. The experiments on vision tasks are limited, and the tested datasets are small.
2. The algorithm may take too long to pick out pruning elements, which is time-inefficiency.
3. Some contributions are over-claimed. For example, it is still convenient for many other algorithms to prune with a predefined budget.

**Justification For Why Not Higher Score:**

The main concerns from reviewers and me are three-fold:
1. The algorithm is time-inefficiency, which may take much time to decide the pruning elements, which is raised all reviewers.
2. More experiments on larger dataset are needed. (By Reviewer #Huai)
3. More pruning algorithms and metrics should be compared to show the effectiveness.

Totally, though the proposed algorithms show some advantages compared with previous ones, the reviewers and I still have concerns about the efficiency of the algorithm and the integrity of experiments.

**Justification For Why Not Lower Score:**

N/A

---

### Decision · Program_Chairs · 2024-01-16

Reject